# Streaming, Distributed Variational Inference for Bayesian Nonparametrics

**Trevor Campbell**[1]   **Julian Straub**[2]   **John W. Fisher III**[2]   **Jonathan P. How**[1]

[1]LIDS, [2]CSAIL, MIT

{tdjc@ , jstraub@csail. , fisher@csail. , jhow@}mit.edu

## Abstract

This paper presents a methodology for creating streaming, distributed inference algorithms for Bayesian nonparametric (BNP) models. In the proposed framework, processing nodes receive a sequence of data minibatches, compute a variational posterior for each, and make asynchronous streaming updates to a central model. In contrast to previous algorithms, the proposed framework is truly streaming, distributed, asynchronous, learning-rate-free, and truncation-free. The key challenge in developing the framework, arising from the fact that BNP models do not impose an inherent ordering on their components, is finding the correspondence between minibatch and central BNP posterior components before performing each update. To address this, the paper develops a combinatorial optimization problem over component correspondences, and provides an efficient solution technique. The paper concludes with an application of the methodology to the DP mixture model, with experimental results demonstrating its practical scalability and performance.

## 1   Introduction

Bayesian nonparametric (BNP) stochastic processes are *streaming* priors – their unique feature is that they specify, in a probabilistic sense, that the complexity of a latent model should grow as the amount of observed data increases. This property captures common sense in many data analysis problems – for example, one would expect to encounter far more topics in a document corpus after reading $10^6$ documents than after reading $10$ – and becomes crucial in settings with unbounded, persistent streams of data. While their fixed, parametric cousins can be used to infer model complexity for datasets with known magnitude a priori [1, 2], such priors are silent with respect to notions of model complexity growth in streaming data settings.

Bayesian nonparametrics are also naturally suited to *parallelization* of data processing, due to the exchangeability, and thus conditional independence, they often exhibit via de Finetti's theorem. For example, labels from the Chinese Restaurant process [3] are rendered i.i.d. by conditioning on the underlying Dirichlet process (DP) random measure, and feature assignments from the Indian Buffet process [4] are rendered i.i.d. by conditioning on the underlying beta process (BP) random measure.

Given these properties, one might expect there to be a wealth of inference algorithms for BNPs that address the challenges associated with parallelization and streaming. However, previous work has only addressed these two settings in concert for parametric models [5, 6], and only recently has each been addressed individually for BNPs. In the streaming setting, [7] and [8] developed streaming inference for DP mixture models using sequential variational approximation. Stochastic variational inference [9] and related methods [10–13] are often considered streaming algorithms, but their performance depends on the choice of a learning rate and on the dataset having known, fixed size a priori [5]. Outside of variational approaches, which are the focus of the present paper, there exist exact parallelized MCMC methods for BNPs [14, 15]; the tradeoff in using such methods is that they provide samples from the posterior rather than the distribution itself, and results regarding assessing

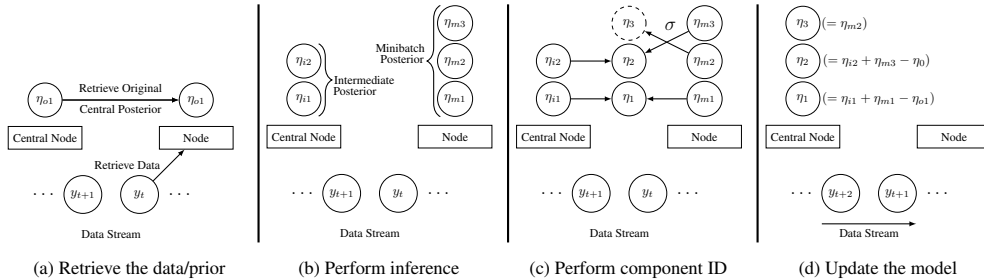

(a) Retrieve the data/prior    (b) Perform inference    (c) Perform component ID    (d) Update the model

Figure 1: The four main steps of the algorithm that is run asynchronously on each processing node.

convergence remain limited. Sequential particle filters for inference have also been developed [16], but these suffer issues with particle degeneracy and exponential forgetting.

The main challenge posed by the streaming, distributed setting for BNPs is the combinatorial problem of *component identification*. Most BNP models contain some notion of a countably infinite set of latent "components" (e.g. clusters in a DP mixture model), and do not impose an inherent ordering on the components. Thus, in order to combine information about the components from multiple processors, the correspondence between components must first be found. Brute force search is intractable even for moderately sized models – there are $\binom{K_1+K_2}{K_1}$ possible correspondences for two sets of components of sizes $K_1$ and $K_2$. Furthermore, there does not yet exist a method to evaluate the quality of a component correspondence for BNP models. This issue has been studied before in the MCMC literature, where it is known as the "label switching problem", but past solution techniques are generally model-specific and restricted to use on very simple mixture models [17, 18].

This paper presents a methodology for creating streaming, distributed inference algorithms for Bayesian nonparametric models. In the proposed framework (shown for a single node A in Figure 1), processing nodes receive a sequence of data minibatches, compute a variational posterior for each, and make asynchronous streaming updates to a central model using a mapping obtained from a component identification optimization. The key contributions of this work are as follows. First, we develop a minibatch posterior decomposition that motivates a learning-rate-free streaming, distributed framework suitable for Bayesian nonparametrics. Then, we derive the component identification optimization problem by maximizing the probability of a component matching. We show that the BNP prior regularizes model complexity in the optimization; an interesting side effect of this is that regardless of whether the minibatch variational inference scheme is truncated, the proposed algorithm is truncation-free. Finally, we provide an efficiently computable regularization bound for the Dirichlet process prior based on Jensen's inequality[1]. The paper concludes with applications of the methodology to the DP mixture model, with experimental results demonstrating the scalability and performance of the method in practice.

## 2 Streaming, distributed Bayesian nonparametric inference

The proposed framework, motivated by a posterior decomposition that will be discussed in Section 2.1, involves a collection of processing nodes with asynchronous access to a central variational posterior approximation (shown for a single node in Figure 1). Data is provided to each processing node as a sequence of minibatches. When a processing node receives a minibatch of data, it obtains the central posterior (Figure 1a), and using it as a prior, computes a minibatch variational posterior approximation (Figure 1b). When minibatch inference is complete, the node then performs component identification between the minibatch posterior and the current central posterior, accounting for possible modifications made by other processing nodes (Figure 1c). Finally, it merges the minibatch posterior into the central variational posterior (Figure 1d).

In the following sections, we use the DP mixture [3] as a guiding example for the technical development of the inference framework. However, it is emphasized that the material in this paper generalizes to many other BNP models, such as the hierarchical DP (HDP) topic model [19], BP latent feature model [20], and Pitman-Yor (PY) mixture [21] (see the supplement for further details).

## 2.1 Posterior decomposition

Consider a DP mixture model [3], with cluster parameters $\theta$, assignments $z$, and observed data $y$. For each asynchronous update made by each processing node, the dataset is split into three subsets $y = y_o \cup y_i \cup y_m$ for analysis. When the processing node receives a **minibatch** of data $y_m$, it queries the central processing node for the **original** posterior $p(\theta, z_o|y_o)$, which will be used as the prior for minibatch inference. Once inference is complete, it again queries the central processing node for the **intermediate** posterior $p(\theta, z_o, z_i|y_o, y_i)$ which accounts for asynchronous updates from other processing nodes since minibatch inference began. Each subset $y_r$, $r \in \{o, i, m\}$, has $N_r$ observations $\{y_{rj}\}_{j=1}^{N_r}$, and each variable $z_{rj} \in \mathbb{N}$ assigns $y_{rj}$ to cluster parameter $\theta_{z_{rj}}$. Given the independence of $\theta$ and $z$ in the prior, and the conditional independence of the data given the latent parameters, Bayes' rule yields the following decomposition of the posterior of $\theta$ and $z$ given $y$,

$$\overbrace{p(\theta, z|y)}^{\text{Updated Central Posterior}} \propto \frac{p(z_i, z_m|z_o)}{p(z_i|z_o)p(z_m|z_o)} \cdot \overbrace{p(\theta, z_o|y_o)^{-1}}^{\text{Original Posterior}} \cdot \overbrace{p(\theta, z_m, z_o|y_m, y_o)}^{\text{Minibatch Posterior}} \cdot \overbrace{p(\theta, z_i, z_o|y_i, y_o)}^{\text{Intermediate Posterior}}. \quad (1)$$

This decomposition suggests a simple streaming, distributed, asynchronous update rule for a processing node: first, obtain the current central posterior density $p(\theta, z_o|y_o)$, and using it as a prior, compute the minibatch posterior $p(\theta, z_m, z_o|y_o, y_m)$; and then update the central posterior density by using (1) with the current central posterior density $p(\theta, z_i, z_o|y_i, y_o)$. However, there are two issues preventing the direct application of the decomposition rule (1):

**Unknown component correspondence:** Since it is generally intractable to find the minibatch posteriors $p(\theta, z_m, z_o|y_o, y_m)$ exactly, approximate methods are required. Further, as (1) requires the multiplication of densities, sampling-based methods are difficult to use, suggesting a variational approach. Typical mean-field variational techniques introduce an artificial ordering of the parameters in the posterior, thereby breaking symmetry that is crucial to combining posteriors correctly using density multiplication [6]. The use of (1) with mean-field variational approximations thus requires first solving a component identification problem.

**Unknown model size:** While previous posterior merging procedures required a 1-to-1 matching between the components of the minibatch posterior and central posterior [5, 6], Bayesian nonparametric posteriors break this assumption. Indeed, the datasets $y_o$, $y_i$, and $y_m$ from the same nonparametric mixture model can be generated by the same, disjoint, or an overlapping set of cluster parameters. In other words, the global number of unique posterior components cannot be determined until the component identification problem is solved and the minibatch posterior is merged.

## 2.2 Variational component identification

Suppose we have the following mean-field exponential family prior and approximate variational posterior densities in the minibatch decomposition (1),

$$p(\theta_k) = h(\theta_k)e^{\eta_0^T T(\theta_k) - A(\eta_0)} \ \forall k \in \mathbb{N}$$

$$p(\theta, z_o|y_o) \simeq q_o(\theta, z_o) = \zeta_o(z_o) \prod_{k=1}^{K_o} h(\theta_k)e^{\eta_{ok}^T T(\theta_k) - A(\eta_{ok})}$$

$$p(\theta, z_m, z_o|y_m, y_o) \simeq q_m(\theta, z_m, z_o) = \zeta_m(z_m)\zeta_o(z_o) \prod_{k=1}^{K_m} h(\theta_k)e^{\eta_{mk}^T T(\theta_k) - A(\eta_{mk})} \quad (2)$$

$$p(\theta, z_i, z_o|y_i, y_o) \simeq q_i(\theta, z_i, z_o) = \zeta_i(z_i)\zeta_o(z_o) \prod_{k=1}^{K_i} h(\theta_k)e^{\eta_{ik}^T T(\theta_k) - A(\eta_{ik})},$$

where $\zeta_r(\cdot)$, $r \in \{o, i, m\}$ are products of categorical distributions for the cluster labels $z_r$, and the goal is to use the posterior decomposition (1) to find the updated posterior approximation

$$p(\theta, z|y) \simeq q(\theta, z) = \zeta(z) \prod_{k=1}^{K} h(\theta_k)e^{\eta_k^T T(\theta_k) - A(\eta_k)}. \quad (3)$$

As mentioned in the previous section, the artificial ordering of components causes the naïve application of (1) with variational approximations to fail, as disparate components from the approximate posteriors may be merged erroneously. This is demonstrated in Figure 3a, which shows results from a synthetic experiment (described in Section 4) ignoring component identification. As the number of parallel threads increases, more matching mistakes are made, leading to decreasing model quality.

To address this, first note that there is no issue with the first $K_o$ components of $q_m$ and $q_i$; these can be merged directly since they each correspond to the $K_o$ components of $q_o$. Thus, the component identification problem reduces to finding the correspondence between the last $K_m' = K_m - K_o$ components of the minibatch posterior and the last $K_i' = K_i - K_o$ components of the intermediate posterior. For notational simplicity (and without loss of generality), fix the component ordering of the intermediate posterior $q_i$, and define $\sigma : [K_m] \to [K_i + K_m']$ to be the 1-to-1 mapping from minibatch posterior component $k$ to updated central posterior component $\sigma(k)$, where $[K] := \{1, \dots, K\}$. The fact that the first $K_o$ components have no ordering ambiguity can be expressed as $\sigma(k) = k \ \forall k \in [K_o]$. Note that the maximum number of components after merging is $K_i + K_m'$, since each of the last $K_m'$ components in the minibatch posterior may correspond to new components in the intermediate posterior. After substituting the three variational approximations (2) into (1), the goal of the component identification optimization is to find the 1-to-1 mapping $\sigma^\star$ that yields the largest updated posterior normalizing constant, i.e. matches components with similar densities,

$$\sigma^\star \leftarrow \operatorname*{argmax}_{\sigma} \quad \sum_z \int_\theta \frac{p(z_i, z_m | z_o)}{p(z_i | z_o) p(z_m | z_o)} q_o(\theta, z_o)^{-1} q_m^\sigma(\theta, z_m, z_o) q_i(\theta, z_i, z_o)$$

$$\text{s.t.} \quad q_m^\sigma(\theta, z_m) = \zeta_m^\sigma(z_m) \prod_{k=1}^{K_m} h(\theta_{\sigma(k)}) e^{\eta_{mk}^T T(\theta_{\sigma(k)}) - A(\eta_{mk})} \tag{4}$$

$$\sigma(k) = k, \ \forall k \in [K_o], \sigma \text{ 1-to-1}$$

where $\zeta_m^\sigma(z_m)$ is the distribution such that $\mathrm{P}_{\zeta_m^\sigma}(z_{mj} = \sigma(k)) = \mathrm{P}_{\zeta_m}(z_{mj} = k)$. Taking the logarithm of the objective and exploiting the mean-field decoupling allows the separation of the objective into a sum of two terms: one expressing the quality of the matching between components (the integral over $\theta$), and one that regularizes the final model size (the sum over $z$). While the first term is available in closed form, the second is in general not. Therefore, using the concavity of the logarithm function, Jensen's inequality yields a lower bound that can be used in place of the intractable original objective, resulting in the final component identification optimization:

$$\sigma^\star \leftarrow \operatorname*{argmax}_{\sigma} \quad \sum_{k=1}^{K_i + K_m'} A\left(\tilde{\eta}_k^\sigma\right) + \mathbb{E}_\zeta^\sigma \left[\log p(z_i, z_m, z_o)\right]$$

$$\text{s.t.} \quad \tilde{\eta}_k^\sigma = \tilde{\eta}_{ik} + \tilde{\eta}_{mk}^\sigma - \tilde{\eta}_{ok} \tag{5}$$

$$\sigma(k) = k \ \forall k \in [K_o], \sigma \text{ 1-to-1.}$$

A more detailed derivation of the optimization may be found in the supplement. $\mathbb{E}_\zeta^\sigma$ denotes expectation under the distribution $\zeta_o(z_o)\zeta_i(z_i)\zeta_m^\sigma(z_m)$, and

$$\tilde{\eta}_{rk} = \begin{cases} \eta_{rk} & k \le K_r \\ \eta_0 & k > K_r \end{cases} \ \forall r \in \{o, i, m\}, \quad \tilde{\eta}_{mk}^\sigma = \begin{cases} \eta_{m\sigma^{-1}(k)} & k \in \sigma([K_m]) \\ \eta_0 & k \notin \sigma([K_m]) \end{cases}, \tag{6}$$

where $\sigma([K_m])$ denotes the range of the mapping $\sigma$. The definitions in (6) ensure that the prior $\eta_0$ is used whenever a posterior $r \in \{i, m, o\}$ does not contain a particular component $k$. The intuition for the optimization (5) is that it combines finding component correspondences with high similarity (via the log-partition function) with a regularization term[2] on the final updated posterior model size.

Despite its motivation from the Dirichlet process mixture, the component identification optimization (5) is not specific to this model. Indeed, the derivation did not rely on any properties specific to the Dirichlet process mixture; the optimization applies to any Bayesian nonparametric model with a set of "components" $\theta$, and a set of combinatorial "indicators" $z$. For example, the optimization applies to the hierarchical Dirichlet process topic model [10] with topic word distributions $\theta$ and local-to-global topic correspondences $z$, and to the beta process latent feature model [4] with features $\theta$ and

binary assignment vectors $z$. The form of the objective in the component identification optimization (5) reflects this generality. In order to apply the proposed streaming, distributed method to a particular model, one simply needs a black-box variational inference algorithm that computes posteriors of the form (2), and a way to compute or bound the expectation in the objective of (5).

## 2.3 Updating the central posterior

To update the central posterior, the node first locks it and solves for $\sigma^\star$ via (5). Locking prevents other nodes from solving (5) or modifying the central posterior, but does not prevent other nodes from reading the central posterior, obtaining minibatches, or performing inference; the synthetic experiment in Section 4 shows that this does not incur a significant time penalty in practice. Then the processing node transmits $\sigma^\star$ and its minibatch variational posterior to the central processing node where the product decomposition (1) is used to find the updated central variational posterior $q$ in (3), with parameters

$$K = \max\left\{ K_i, \max_{k \in [K_m]} \sigma^\star(k) \right\}, \quad \zeta(z) = \zeta_i(z_i)\zeta_o(z_o)\zeta_m^{\sigma^\star}(z_m), \quad \eta_k = \tilde{\eta}_{ik} + \tilde{\eta}_{mk}^{\sigma^\star} - \tilde{\eta}_{ok}. \quad (7)$$

Finally, the node unlocks the central posterior, and the next processing node to receive a new minibatch will use the above $K$, $\zeta(z)$, and $\eta_k$ from the central node as their $K_o$, $\zeta_o(z_o)$, and $\eta_{ok}$.

# 3 Application to the Dirichlet process mixture model

The expectation in the objective of (5) is typically intractable to compute in closed-form; therefore, a suitable lower bound may be used in its place. This section presents such a bound for the Dirichlet process, and discusses the application of the proposed inference framework to the Dirichlet process mixture model using the developed bound. Crucially, the lower bound decomposes such that the optimization (5) becomes a maximum-weight bipartite matching problem. Such problems are solvable in polynomial time [22] by the Hungarian algorithm, leading to a tractable component identification step in the proposed streaming, distributed framework.

## 3.1 Regularization lower bound

For the Dirichlet process with concentration parameter $\alpha > 0$, $p(z_i, z_m, z_o)$ is the Exchangeable Partition Probability Function (EPPF) [23]

$$p(z_i, z_m, z_o) \propto \alpha^{|\mathcal{K}|-1} \prod_{k \in \mathcal{K}} (n_k - 1)!, \quad (8)$$

where $n_k$ is the amount of data assigned to cluster $k$, and $\mathcal{K}$ is the set of labels of nonempty clusters. Given that the variational distribution $\zeta_r(z_r)$, $r \in \{i, m, o\}$ is a product of independent categorical distributions $\zeta_r(z_r) = \prod_{j=1}^{N_r} \prod_{k=1}^{K_r} \pi_{rjk}^{\mathbb{1}[z_{rj}=k]}$, Jensen's inequality may be used to bound the regularization in (5) below (see the supplement for further details) by

$$\mathbb{E}_\zeta^\sigma \left[ \log p(z_i, z_m, z_o) \right] \geq \sum_{k=1}^{K_i + K'_m} \left( 1 - e^{\tilde{s}_k^\sigma} \right) \log \alpha + \log \Gamma \left( \max\left\{ 2, \tilde{t}_k^\sigma \right\} \right) + C \quad (9)$$

$$\tilde{s}_k^\sigma = \tilde{s}_{ik} + \tilde{s}_{mk}^\sigma + \tilde{s}_{ok}, \quad \tilde{t}_k^\sigma = \tilde{t}_{ik} + \tilde{t}_{mk}^\sigma + \tilde{t}_{ok},$$

where $C$ is a constant with respect to the component mapping $\sigma$, and

$$\tilde{s}_{rk} = \begin{cases} \sum_{j=1}^{N_r} \log(1 - \pi_{rjk}) & k \leq K_r \\ 0 & k > K_r \end{cases} \forall r \in \{o, i, m\} \quad \tilde{t}_{rk} = \begin{cases} \sum_{j=1}^{N_r} \pi_{rjk} & k \leq K_r \\ 0 & k > K_r \end{cases} \forall r \in \{o, i, m\}$$

$$\tilde{s}_{mk}^\sigma = \begin{cases} \sum_{j=1}^{N_m} \log(1 - \pi_{mj\sigma^{-1}(k)}) & k \in \sigma([K_m]) \\ 0 & k \notin \sigma([K_m]) \end{cases} \quad \tilde{t}_{mk}^\sigma = \begin{cases} \sum_{j=1}^{N_m} \pi_{mj\sigma^{-1}(k)} & k \in \sigma([K_m]) \\ 0 & k \notin \sigma([K_m]) \end{cases}. \quad (10)$$

Note that the bound (9) allows incremental updates: after finding the optimal mapping $\sigma^\star$, the central update (7) can be augmented by updating the values of $s_k$ and $t_k$ on the central node to

$$s_k \leftarrow \tilde{s}_{ik} + \tilde{s}_{mk}^{\sigma^\star} + \tilde{s}_{ok}, \quad t_k \leftarrow \tilde{t}_{ik} + \tilde{t}_{mk}^{\sigma^\star} + \tilde{t}_{ok}. \quad (11)$$

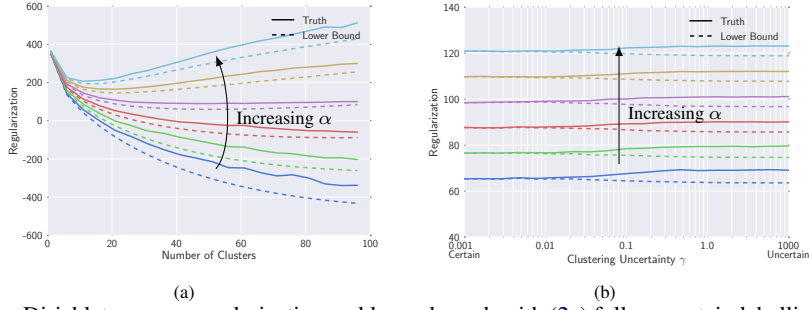

(a)                                                                 (b)

Figure 2: The Dirichlet process regularization and lower bound, with (2a) fully uncertain labelling and varying number of clusters, and (2b) the number of clusters fixed with varying labelling uncertainty.

As with $K$, $\eta_k$, and $\zeta$ from (7), after performing the regularization statistics update (11), a processing node that receives a new minibatch will use the above $s_k$ and $t_k$ as their $s_{ok}$ and $t_{ok}$, respectively.

Figure 2 demonstrates the behavior of the lower bound in a synthetic experiment with $N = 100$ datapoints for various DP concentration parameter values $\alpha \in \left[10^{-3}, 10^3\right]$. The true regularization $\log \mathbb{E}_\zeta \left[p(z)\right]$ was computed by sample approximation with $10^4$ samples. In Figure 2a, the number of clusters $K$ was varied, with symmetric categorical label weights set to $\frac{1}{K}$. This figure demonstrates two important phenomena. First, the bound increases as $K \to 0$; in other words, it gives preference to fewer, larger clusters, which is the typical BNP "rich get richer" property. Second, the behavior of the bound as $K \to N$ depends on the concentration parameter $\alpha$ – as $\alpha$ increases, more clusters are preferred. In Figure 2b, the number of clusters $K$ was fixed to 10, and the categorical label weights were sampled from a symmetric Dirichlet distribution with parameter $\gamma \in \left[10^{-3}, 10^3\right]$. This figure demonstrates that the bound does not degrade significantly with high labelling uncertainty, and is nearly exact for low labelling uncertainty. Overall, Figure 2a demonstrates that the proposed lower bound exhibits similar behaviors to the true regularization, supporting its use in the optimization (5).

## 3.2 Solving the component identification optimization

Given that both the regularization (9) and component matching score in the objective (5) decompose as a sum of terms for each $k \in [K_i + K_m']$, the objective can be rewritten using a matrix of matching scores $\mathbf{R} \in \mathbb{R}^{\left(K_i + K_m'\right) \times \left(K_i + K_m'\right)}$ and selector variables $\mathbf{X} \in \{0,1\}^{\left(K_i + K_m'\right) \times \left(K_i + K_m'\right)}$. Setting $\mathbf{X}_{kj} = 1$ indicates that component $k$ in the minibatch posterior is matched to component $j$ in the intermediate posterior (i.e. $\sigma(k) = j$), providing a score $\mathbf{R}_{kj}$ defined using (6) and (10) as

$$\mathbf{R}_{kj} = A\left(\tilde{\eta}_{ij} + \tilde{\eta}_{mk} - \tilde{\eta}_{oj}\right) + \left(1 - e^{\tilde{s}_{ij} + \tilde{s}_{mk} + \tilde{s}_{oj}}\right)\log \alpha + \log \Gamma\left(\max\left\{2, \tilde{t}_{ij} + \tilde{t}_{mk} + \tilde{t}_{oj}\right\}\right). \quad (12)$$

The optimization (5) can be rewritten in terms of $\mathbf{X}$ and $\mathbf{R}$ as

$$\mathbf{X}^\star \leftarrow \underset{\mathbf{X}}{\text{argmax}} \quad \text{tr}\left[\mathbf{X}^T \mathbf{R}\right]$$
$$\text{s.t.} \quad \mathbf{X}\mathbf{1} = \mathbf{1}, \quad \mathbf{X}^T\mathbf{1} = \mathbf{1}, \quad \mathbf{X}_{kk} = 1, \forall k \in [K_o] \quad (13)$$
$$\mathbf{X} \in \{0,1\}^{\left(K_i + K_m'\right) \times \left(K_i + K_m'\right)}, \quad \mathbf{1} = [1, \dots, 1]^T.$$

The first two constraints express the 1-to-1 property of $\sigma(\cdot)$. The constraint $\mathbf{X}_{kk} = 1 \forall k \in [K_o]$ fixes the upper $K_o \times K_o$ block of $\mathbf{X}$ to $\mathbf{I}$ (due to the fact that the first $K_o$ components are matched directly), and the off-diagonal blocks to $\mathbf{0}$. Denoting $\mathbf{X}'$, $\mathbf{R}'$ to be the lower right $\left(K_i' + K_m'\right) \times \left(K_i' + K_m'\right)$ blocks of $\mathbf{X}$, $\mathbf{R}$, the remaining optimization problem is a linear assignment problem on $\mathbf{X}'$ with cost matrix $-\mathbf{R}'$, which can be solved using the Hungarian algorithm[3]. Note that if $K_m = K_o$ or $K_i = K_o$, this implies that no matching problem needs to be solved – the first $K_o$ components of the minibatch posterior are matched directly, and the last $K_m'$ are set as new components. In practical implementation of the framework, new clusters are typically discovered at a diminishing rate as more data are observed, so the number of matching problems that are solved likewise tapers off. The final optimal component mapping $\sigma^\star$ is found by finding the nonzero elements of $\mathbf{X}^\star$:

$$\sigma^\star(k) \leftarrow \underset{j}{\text{argmax}} \, \mathbf{X}_{kj}^\star \, \forall k \in [K_m]. \quad (14)$$

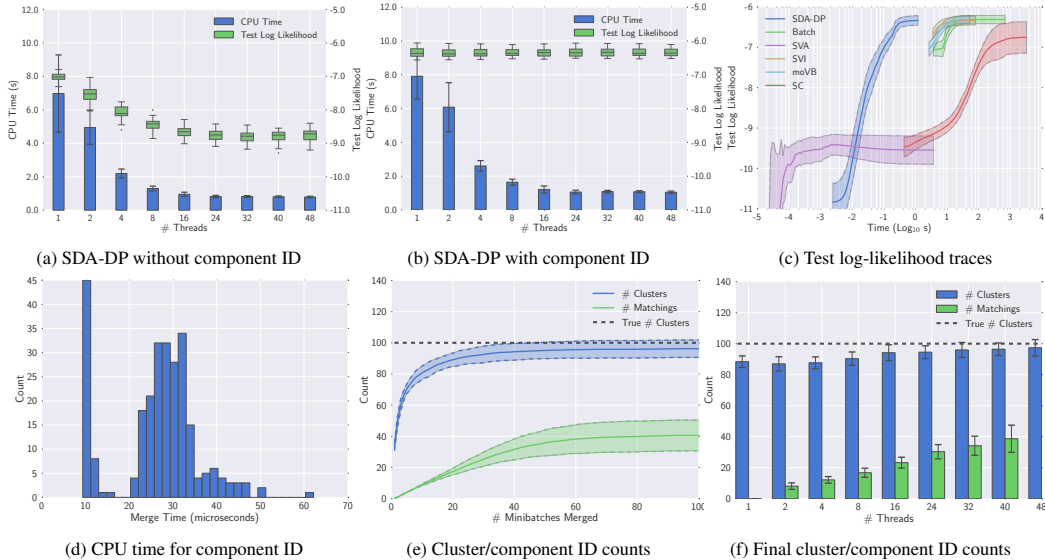

Figure 3: Synthetic results over 30 trials. (3a-3b) Computation time and test log likelihood for SDA-DP with varying numbers of parallel threads, with component identification disabled (3a) and enabled (3b). (3c) Test log likelihood traces for SDA-DP (40 threads) and the comparison algorithms. (3d) Histogram of computation time (in microseconds) to solve the component identification optimization. (3e) Number of clusters and number of component identification problems solved as a function of the number of minibatch updates (40 threads). (3f) Final number of clusters and matchings solved with varying numbers of parallel threads.

## 4 Experiments

In this section, the proposed inference framework is evaluated on the DP Gaussian mixture with a normal-inverse-Wishart (NIW) prior. We compare the streaming, distributed procedure coupled with standard variational inference [24] (SDA-DP) to five state-of-the-art inference algorithms: memoized online variational inference (moVB) [13], stochastic online variational inference (SVI) [9] with learning rate $(t+10)^{-\frac{1}{2}}$, sequential variational approximation (SVA) [7] with cluster creation threshold $10^{-1}$ and prune/merge threshold $10^{-3}$, subcluster splits MCMC (SC) [14], and batch variational inference (Batch) [24]. Priors were set by hand and all methods were initialized randomly. Methods that use multiple passes through the data (e.g. moVB, SVI) were allowed to do so. moVB was allowed to make birth/death moves, while SVI/Batch had fixed truncations. All experiments were performed on a computer with 24 CPU cores and 12GiB of RAM.

**Synthetic:** This dataset consisted of 100,000 2-dimensional vectors generated from a Gaussian mixture model with 100 clusters and a $\mathrm{NIW}(\mu_0, \kappa_0, \Psi_0, \nu_0)$ prior with $\mu_0 = 0$, $\kappa_0 = 10^{-3}$, $\Psi_0 = I$, and $\nu_0 = 4$. The algorithms were given the true NIW prior, DP concentration $\alpha = 5$, and minibatches of size 50. SDA-DP minibatch inference was truncated to $K = 50$ components, and all other algorithms were truncated to $K = 200$ components. Figure 3 shows the results from the experiment over 30 trials, which illustrate a number of important properties of SDA-DP. First and foremost, ignoring the component identification problem leads to decreasing model quality with increasing number of parallel threads, since more matching mistakes are made (Figure 3a). Second, if component identification is properly accounted for using the proposed optimization, increasing the number of parallel threads reduces execution time, but does not affect the final model quality (Figure 3b). Third, SDA-DP (with 40 threads) converges to the same final test log likelihood as the comparison algorithms in significantly reduced time (Figure 3c). Fourth, each component identification optimization typically takes $\sim 10^{-5}$ seconds, and thus matching accounts for less than a millisecond of total computation and does not affect the overall computation time significantly (Figure 3d). Fifth, the majority of the component matching problems are solved within the first 80 minibatch updates (out of a total of 2,000) – afterwards, the true clusters have all been discovered and the processing nodes contribute to those clusters rather than creating new ones, as per the discussion at the end of Section 3.2 (Figure 3e). Finally, increased parallelization can be advantageous in discovering the correct number of clusters; with only one thread, mistakes made early on are built upon and persist, whereas with more threads there are more component identification problems solved, and thus more chances to discover the correct clusters (Figure 3f).

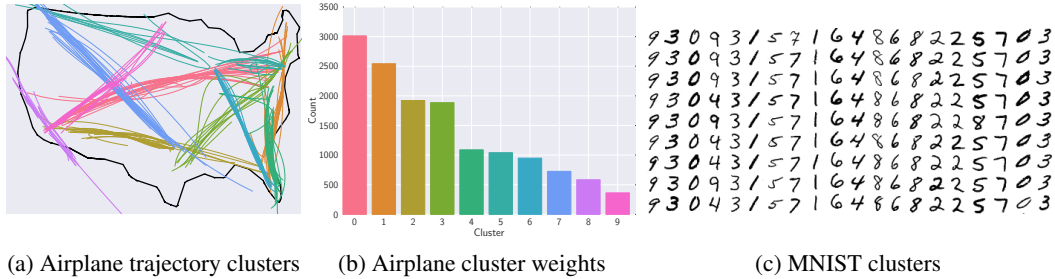

| (a) Airplane trajectory clusters | (b) Airplane cluster weights | (c) MNIST clusters |

(d) Numerical results on Airplane, MNIST, and SUN

| Algorithm | Airplane Time (s) | Airplane TestLL | MNIST Time (s) | MNIST TestLL | SUN Time (s) | SUN TestLL |
|---|---|---|---|---|---|---|
| SDA-DP | 0.66 | -0.55 | 3.0 | -145.3 | 9.4 | -150.3 |
| SVI | 1.50 | -0.59 | 117.4 | -147.1 | 568.9 | -149.9 |
| SVA | 3.00 | -4.71 | 57.0 | -145.0 | 10.4 | -152.8 |
| moVB | 0.69 | -0.72 | 645.9 | -149.2 | 1258.1 | -149.7 |
| SC | 393.6 | -1.06 | 1639.1 | -146.8 | 1618.4 | -150.6 |
| Batch | 1.07 | -0.72 | 829.6 | -149.5 | 1881.5 | -149.7 |

Figure 4: (4a-4b) Highest-probability instances and counts for 10 trajectory clusters generated by SDA-DP. (4c) Highest-probability instances for 20 clusters discovered by SDA-DP on MNIST. (4d) Numerical results.

**Airplane Trajectories:** This dataset consisted of ∼3,000,000 automatic dependent surveillance broadcast (ADS-B) messages collected from planes across the United States during the period 2013-03-22 01:30:00UTC to 2013-03-28 12:00:00UTC. The messages were connected based on plane call sign and time stamp, and erroneous trajectories were filtered based on reasonable spatial/temporal bounds, yielding 15,022 trajectories with 1,000 held out for testing. The latitude/longitude points in each trajectory were fit via linear regression, and the 3-dimensional parameter vectors were clustered. Data was split into minibatches of size 100, and SDA-DP used 16 parallel threads.

**MNIST Digits [25]:** This dataset consisted of 70,000 $28 \times 28$ images of hand-written digits, with 10,000 held out for testing. The images were reduced to 20 dimensions with PCA prior to clustering. Data was split into minibatches of size 500, and SDA-DP used 48 parallel threads.

**SUN Images [26]:** This dataset consisted of 108,755 images from 397 scene categories, with 8,755 held out for testing. The images were reduced to 20 dimensions with PCA prior to clustering. Data was split into minibatches of size 500, and SDA-DP used 48 parallel threads.

Figure 4 shows the results from the experiments on the three real datasets. From a qualitative standpoint, SDA-DP discovers sensible clusters in the data, as demonstrated in Figures 4a–4c. However, an important quantitative result is highlighted by Table 4d: the larger a dataset is, the more the benefits of parallelism provided by SDA-DP become apparent. SDA-DP consistently provides a model quality that is competitive with the other algorithms, but requires orders of magnitude less computation time, corroborating similar findings on the synthetic dataset.

## 5   Conclusions

This paper presented a streaming, distributed, asynchronous inference algorithm for Bayesian nonparametric models, with a focus on the combinatorial problem of matching minibatch posterior components to central posterior components during asynchronous updates. The main contributions are a component identification optimization based on a minibatch posterior decomposition, a tractable bound on the objective for the Dirichlet process mixture, and experiments demonstrating the performance of the methodology on large-scale datasets. While the present work focused on the DP mixture as a guiding example, it is not limited to this model – exploring the application of the proposed methodology to other BNP models is a potential area for future research.

### Acknowledgments

This work was supported by the Office of Naval Research under ONR MURI grant N000141110688.

## Footnotes

[1]Regularization bounds for other popular BNP priors may be found in the supplement.

[2]This is equivalent to the KL-divergence regularization $-\mathrm{KL}\left[\zeta_o(z_o)\zeta_i(z_i)\zeta_m^\sigma(z_m) \, \middle|\middle| \, p(z_i, z_m, z_o)\right]$.

[3] For the experiments in this work, we used the implementation at github.com/hrldcpr/hungarian.

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
