[Supplementary Material]

# Streaming, Distributed Variational Inference for Bayesian Nonparametrics – Supplement

**Trevor Campbell**[1]     **Julian Straub**[2]     **John W. Fisher III**[2]     **Jonathan P. How**[1]
[1]LIDS, [2]CSAIL, MIT
{tdjc@ , jstraub@csail. , fisher@csail. , jhow@}mit.edu

This document covers a more detailed derivation of the merge optimization in Section 1, a more detailed derivation of the regularization bound for the Dirichlet process in Section 2, and finally a set of regularization bounds for other nonparametric processes in Section 3.

## 1  Merge optimization derivation

In the main text, the merge optimization is derived with a number of steps withheld for brevity. Here, the derivation is presented more fully. First, recall the maximization of the posterior normalization constant with variational posteriors substituted for the true posteriors:

$$
\sigma^\star \leftarrow \arg\max_\sigma \quad \sum_z \int_\theta \frac{p(z_i, z_m | z_o)}{p(z_i | z_o) p(z_m | z_o)} q_o(\theta, z_o)^{-1} q_m^\sigma(\theta, z_m, z_o) q_i(\theta, z_i, z_o)
$$

$$
\text{s.t.} \quad q_m^\sigma(\theta, z_m) = \zeta_m^\sigma(z_m) \prod_{k=1}^{K_m} h(\theta_{\sigma(k)}) e^{\eta_{mk}^T T(\theta_{\sigma(k)}) - A(\eta_{mk})}
$$

$$
\sigma(k) = k, \ \forall k \in [K_o], \sigma \text{ 1-to-1.}
$$

(1)

Taking the logarithm of the objective does not change $\sigma^\star$ since it is strictly monotonic, and exploiting the decoupling of $\theta$ and $z$ in $q_o$, $q_m^\sigma$, and $q_i$ yields an objective split into two terms:

$$
\sigma^\star \leftarrow \arg\max_\sigma \quad \log \sum_z \frac{p(z_i, z_m | z_o)}{p(z_i | z_o) p(z_m | z_o)} \zeta_o(z_o) \zeta_m^\sigma(z_m) \zeta_i(z_i) + \log \int_\theta \prod_{k=1}^{K_i + K_m'} h(\theta_k) e^{\tilde{\eta}_k^{\sigma T} T(\theta_k)} + C
$$

$$
\text{s.t.} \quad \sigma(k) = k, \ \forall k \in [K_o], \sigma \text{ 1-to-1.}
$$

(2)

Examining the $z$ term first, note that

$$
\log \sum_z \frac{p(z_i, z_m | z_o)}{p(z_i | z_o) p(z_m | z_o)} \zeta_o(z_o) \zeta_m^\sigma(z_m) \zeta_i(z_i) = \log \mathbb{E}_\zeta^\sigma \left[ \frac{p(z_i, z_m | z_o)}{p(z_i | z_o) p(z_m | z_o)} \right].
$$

(3)

Computing this expectation in closed-form is generally intractable, and therefore even evaluating the objective above (let alone optimizing it) is intractable. However, since the logarithm function is concave, Jensen's inequality provides a lower bound:

$$
\log \mathbb{E}_\zeta^\sigma \left[ \frac{p(z_i, z_m | z_o)}{p(z_i | z_o) p(z_m | z_o)} \right] \geq \mathbb{E}_\zeta^\sigma \left[ \log \frac{p(z_i, z_m | z_o)}{p(z_i | z_o) p(z_m | z_o)} \right]
$$

$$
= \mathbb{E}_\zeta^\sigma \left[ \log p(z_i, z_m | z_o) \right] - \mathbb{E}_\zeta^\sigma \left[ \log p(z_i | z_o) \right] - \mathbb{E}_\zeta^\sigma \left[ \log p(z_m | z_o) \right].
$$

(4)

Next, note that $\mathbb{E}_\zeta^\sigma \left[ \log p(z_i | z_o) \right]$ is a constant with respect to $\sigma$ since $\log p(z_i | z_o)$ does not involve any $z_m$ terms. Furthermore, $\mathbb{E}_\zeta^\sigma \left[ \log p(z_m | z_o) \right]$ is invariant with respect to $\sigma$ since Bayesian nonparametric priors are invariant to component relabelling, and conditioning on $z_o$ doesn't affect this property because $\sigma(k) = k \forall k \in [K_o]$. Therefore

$$
\log \mathbb{E}_\zeta^\sigma \left[ \frac{p(z_i, z_m | z_o)}{p(z_i | z_o) p(z_m | z_o)} \right] \geq \mathbb{E}_\zeta^\sigma \left[ \log p(z_i, z_m | z_o) \right] + C.
$$

(5)

Finally adding and subtracting $\mathbb{E}_\zeta^\sigma [\log p(z_o)]$, which is also constant with respect to $\sigma$, we have the final form of the general regularization bound

$$\log \mathbb{E}_\zeta^\sigma \left[ \frac{p(z_i, z_m | z_o)}{p(z_i | z_o) p(z_m | z_o)} \right] \geq \mathbb{E}_\zeta^\sigma [\log p(z_i, z_m, z_o)] + C. \tag{6}$$

Next, examining the $\theta$ term,

$$\log \int_\theta \prod_{k=1}^{K_i + K'_m} h(\theta_k) e^{\tilde{\eta}_k^{\sigma T} T(\theta_k)} = \log \prod_{k=1}^{K_i + K'_m} \int_{\theta_k} h(\theta_k) e^{\tilde{\eta}_k^{\sigma T} T(\theta_k)}$$

$$= \log \prod_{k=1}^{K_i + K'_m} e^{A(\tilde{\eta}_k^\sigma)} \tag{7}$$

$$= \sum_{k=1}^{K_i + K'_m} A\left( \tilde{\eta}_k^\sigma \right).$$

Combining the terms from (6) and (7) yields the optimization shown in the main paper.

## 2  DP regularization derivation

In the main text, the regularization bound for the Dirichlet process prior is derived with some steps withheld for brevity. Here the derivation is presented more fully. For the Dirichlet process with concentration parameter $\alpha > 0$, $p(z_i, z_m, z_o)$ is the Exchangeable Partition Probability Function (EPPF) [Pitman, 1995]

$$p(z_i, z_m, z_o) \propto \alpha^{|\mathcal{K}| - 1} \prod_{k \in \mathcal{K}} (n_k - 1)!, \tag{8}$$

where $n_k$ is the amount of data assigned to cluster $k$, and $\mathcal{K}$ is the set of labels of nonempty clusters,

$$n_k = \sum_{r \in \{i,m,o\}} \sum_{j=1}^{N_r} \mathbb{1}\left[ z_{rj} = k \right], \quad \mathcal{K} = \{ k \in \mathbb{Z} : n_k > 0 \}. \tag{9}$$

Taking the expectation under $\zeta_o(z_o) \zeta_i(z_i) \zeta_m^\sigma(z_m)$,

$$\mathbb{E}_\zeta^\sigma [\log p(z_i, z_m, z_o)] = \mathbb{E}_\zeta^\sigma [|\mathcal{K}|] \log \alpha + \sum_{k \in \mathcal{K}} \mathbb{E}_\zeta^\sigma [\log(n_k - 1)!] + C \tag{10}$$

$$= \mathbb{E}_\zeta^\sigma [|\mathcal{K}|] \log \alpha + \sum_{k \in \mathcal{K}} \mathbb{E}_\zeta^\sigma [\log \Gamma (\max\{2, n_k\})] + C \tag{11}$$

The second equality follows because $\log(n_k - 1)! = \log \Gamma (n_k)$ for all integers $n_k > 0$, and because $\log \Gamma (1) = \log \Gamma (2) = 0$. Expanding $|\mathcal{K}|$ and using the convexity of $\log \Gamma (\max\{2, \cdot\})$,

$$\mathbb{E}_\zeta^\sigma [\log p(z_i, z_m, z_o)] = \sum_{k=1}^K \mathbb{E}_\zeta^\sigma [\mathbb{1}[n_k > 0]] \log \alpha + \sum_{k=1}^K \mathbb{E}_\zeta^\sigma [\log \Gamma (\max\{2, n_k\})] + C \tag{12}$$

$$\geq \sum_{k=1}^K \mathbb{E}_\zeta^\sigma [\mathbb{1}[n_k > 0]] \log \alpha + \sum_{k=1}^K \log \Gamma \left( \max\{2, \mathbb{E}_\zeta^\sigma [n_k]\} \right) + C. \tag{13}$$

Next, the two expectation terms are analyzed given that the variational distribution $\zeta_r(z_r)$, $r \in \{i, m, o\}$ is a product of independent categorical distributions $\zeta_r(z_r) = \prod_{j=1}^{N_r} \prod_{k=1}^{K_r} \pi_{rjk}^{\mathbb{1}[z_{rj}=k]}$. First, the indicator expectation is expressed as

$$\mathbb{E}_\zeta^\sigma [\mathbb{1}[n_k > 0]] = \mathbb{P}(n_k > 0) \tag{14}$$

$$= 1 - \mathbb{P}(n_k = 0) \tag{15}$$

$$= 1 - \mathbb{P}(z_{rj} \neq k \forall r, j) \tag{16}$$

and by the independence of the categorical distributions,

$$\log \mathbb{P}\left(z_{rj} \neq k \forall r, j\right) = \left(\sum_{r \in \{o,i\}} \sum_{j=1}^{N_r} \log(1 - \pi_{rjk})\right) \sum_{j=1}^{N_m} \log(1 - \pi_{mj\sigma^{-1}(k)}) \tag{17}$$

Defining $\tilde{s}_{rk}$, $r \in \{o, i, m\}$ and $\tilde{s}_{mk}^{\sigma}$ as in the paper, we have

$$\log \mathbb{P}\left(z_{rj} \neq k \forall r, j\right) = \tilde{s}_k = \tilde{s}_{ik} + \tilde{s}_{ok} + \tilde{s}_{mk}^{\sigma} \tag{18}$$

and thus

$$\mathbb{E}_{\zeta}^{\sigma}\left[\mathbb{1}\left[n_k > 0\right]\right] = 1 - \mathbb{P}\left(z_{rj} \neq k \forall r, j\right) = 1 - e^{\tilde{s}_k^{\sigma}}. \tag{19}$$

Next, the expectation of the number of observations in cluster $k$ is expressed as

$$\mathbb{E}_{\zeta}^{\sigma}\left[n_k\right] = \mathbb{E}_{\zeta}^{\sigma}\left[\sum_{r \in \{o,i,m\}} \sum_{j=1}^{N_r} \mathbb{1}\left[z_{rj} = k\right]\right] \tag{20}$$

$$= \sum_{r \in \{o,i,m\}} \sum_{j=1}^{N_r} \mathbb{E}_{\zeta}^{\sigma}\left[\mathbb{1}\left[z_{rj} = k\right]\right] \tag{21}$$

$$= \sum_{r \in \{o,i\}} \sum_{j=1}^{N_r} \pi_{rjk} + \sum_{j=1}^{N_m} \pi_{rj\sigma^{-1}(k)} \tag{22}$$

Defining $\tilde{t}_{rk}$, $r \in \{o, i, m\}$ and $\tilde{t}_{mk}^{\sigma}$ as in the paper, we have

$$\mathbb{E}_{\zeta}^{\sigma}\left[n_k\right] = \tilde{t}_k^{\sigma} = \tilde{t}_{ik} + \tilde{t}_{ok} + \tilde{t}_{mk}^{\sigma}. \tag{23}$$

Substituting these two expressions into the lower bound of $\mathbb{E}_{\zeta}^{\sigma}\left[\log p(z_i, z_m, z_o)\right]$ yields the expression presented in the paper.

## 3 Regularization lower bounds for other nonparametric processes

### 3.1 Pitman-Yor Process

For the Pitman-Yor process with concentration parameter $\alpha$ and discount parameter $\gamma$, $p(z)$ is a generalized EPPF [Pitman, 1995]

$$p(z) \propto \gamma^{|\mathcal{K}|-1} \Gamma\left(\frac{\alpha}{\gamma} + |\mathcal{K}|\right) \prod_{k \in \mathcal{K}} (n_k - 1)!. \tag{24}$$

As in the paper $\zeta_r(z_r)$, $r \in \{o, i, m\}$ is assumed to be a product of categorical distributions. In this case, the regularization term can be bounded below by Jensen's inequality using the log-convexity of the gamma function:

$$\mathbb{E}_{\zeta}^{\sigma}\left[\log p(z_i, z_m, z_o)\right]$$

$$\geq \log \Gamma\left(\frac{\alpha}{\gamma} + \sum_{k=1}^{K_i+K_m'}\left(1 - e^{\tilde{s}_k^{\sigma}}\right)\right) + \sum_{k=1}^{K_i+K_m'}\left(1 - e^{\tilde{s}_k^{\sigma}}\right)\log \gamma + \log \Gamma\left(\max\left\{2, \tilde{t}_k^{\sigma}\right\}\right) + C$$

$$\geq \sum_{k=1}^{K_i+K_m'} \log \Gamma\left(1 - e^{\tilde{s}_k^{\sigma}}\right) + \left(1 - e^{\tilde{s}_k^{\sigma}}\right)\log \gamma + \log \Gamma\left(\max\left\{2, \tilde{t}_k^{\sigma}\right\}\right) + C$$

$$\tag{25}$$

where $\tilde{s}_k^{\sigma}$ and $\tilde{t}_k^{\sigma}$ are as specified in the main text, and thus $s_k$ and $t_k$ possess the same efficient update scheme as described in the main text. Further, note that this lower bound is a sum over terms $k \in [K_i + K_m']$; therefore, similarly to the case of the Dirichlet process regularization, the component identification optimization can be solved efficiently via the Hungarian algorithm.

## 3.2 Hierarchical Dirichlet Process

For Sethuraman's stick-breaking construction of the hierarchical Dirichlet process [Wang et al., 2011] with parent concentration $\gamma$ and child concentrations $\alpha_0$, there are two latent labelling variables $c, z$. For $r \in \{o, i, m\}$, $z_{rjl} \in \mathbb{N}$ denotes the atom in the parent Dirichlet process that is linked to atom $l$ in child Dirichlet process $j$, and $c_{rjw} \in \mathbb{N}$ denotes the atom in child Dirichlet process $j$ that is responsible for observation $w$. In each dataset $r \in \{o, i, m\}$ there are $N_r$ child Dirichlet processes $j$ with $N_{rj}$ observations and $T_{rj}$ observed atoms. Given these definitions, $z$ and $c$ are independent and both follow the EPPF for the Dirichlet process:

$$
p(z, c) \propto
$$

$$
\gamma^{|\mathcal{K}|-1} \prod_{k \in \mathcal{K}} (n_k - 1)! \prod_{r \in \{o,i,m\}} \prod_{j=1}^{N_r} \left[ \alpha_0^{T_{rj}-1} \prod_{l=1}^{T_{rj}} (n_{rjl} - 1)! \right] \tag{26}
$$

where $n_k = \sum_{r \in \{o,i,m\}} \sum_{j=1}^{N_r} \sum_{l=1}^{T_{rj}} \mathbb{1}\left[z_{rjl} = k\right]$, $\mathcal{K} = \{k \in \mathbb{Z} : n_k > 0\}$, and $n_{rjl} = \sum_{w=1}^{N_{rj}} \mathbb{1}\left[c_{rjw} = l\right]$. Note, however, that all the terms specific to a particular child Dirichlet process $j$ are constant with respect to reordering the global topics; hence,

$$
\mathbb{E}_\zeta^\sigma \left[\log p(z_i, z_m, z_o, c_i, c_m, c_o)\right] \geq \sum_{k=1}^{K_i + K'_m} \left(1 - e^{\tilde{s}_k^\sigma}\right) \log \alpha + \log \Gamma \left(\max\left\{2, \tilde{t}_k^\sigma\right\}\right) + C \tag{27}
$$

where $\tilde{s}_k^\sigma$ and $\tilde{t}_k^\sigma$ are as specified in the main text, using the categorical distributions on the correspondences between document- and corpus-level topics $z$. Note that $s_k$ and $t_k$ possess the same efficient update scheme as described in the main text. Further, note that this lower bound is a sum over terms $k \in [K_i + K'_m]$; therefore, similarly to the case of the Dirichlet process regularization, the component identification optimization can be solved efficiently via the Hungarian algorithm.

## 3.3 Beta Process

For the 3-parameter beta process with mass parameter $\beta > 0$, concentration parameter $\alpha > 0$, and discount $\gamma \in (0, 1)$, the variable $z$ is a set of indicator variables that denote a feature allocation of the data, where $z_{rjk} = 1$ denotes that observation $j$ in data subset $r \in \{o, i, m\}$ expresses feature $k$, and $z_{rjk} = 0$ otherwise. The function $p(z)$ is the Exchangeable Feature Probability Function (EPPF) [Broderick et al., 2013]

$$
p(z) \propto \frac{1}{|\mathcal{K}|!} \left(\beta \frac{\Gamma(\alpha + 1)}{\Gamma(\alpha + \gamma)}\right)^{|\mathcal{K}|} \prod_{k \in \mathcal{K}} \frac{\Gamma(n_k - \gamma)\Gamma(\alpha + N - n_k + \gamma)}{\Gamma(1 - \gamma)\Gamma(\alpha + N)} \tag{28}
$$

where $n_k = \sum_{r \in \{o,i,m\}} \sum_{j=1}^{N_r} z_{rjk}$, and $\mathcal{K} = \{k \in \mathbb{Z} : n_k > 0\}$. Assuming that $\zeta_r$, $r \in \{o, i, m\}$ are products of independent Bernoulli distributions with parameters $\omega_{rjk} \in [0, 1]$,

$$
\mathbb{E}_\zeta^\sigma \left[\log p(z_i, z_o, z_m)\right]
$$

$$
\tilde{\geq} - \log \Gamma \left(1 + \sum_{k=1}^{K_i + K'_m} \left(1 - e^{\tilde{s}_k^\sigma}\right)\right)
$$

$$
+ \sum_{k=1}^{K_i + K'_m} \left(1 - e^{\tilde{s}_k^\sigma}\right) \log \xi + \log \Gamma \left(\max\{2, \tilde{t}_k^\sigma - \gamma\}\right) + \log \Gamma \left(\max\{2, \alpha + N - \tilde{t}_k^\sigma + \gamma\}\right) + C, \tag{29}
$$

where

$$\xi = \frac{\beta\Gamma\left(\alpha+1\right)}{\Gamma\left(\alpha+\gamma\right)\Gamma\left(1-\gamma\right)\Gamma\left(\alpha+N\right)}$$

$$\tilde{t}_k^\sigma = \tilde{t}_{ik} + \tilde{t}_{ok} + \tilde{t}_{mk}^\sigma$$

$$\tilde{s}_k^\sigma = \tilde{s}_{ik} + \tilde{s}_{ok} + \tilde{s}_{mk}^\sigma$$

$$\tilde{s}_{rk} = \begin{cases} \sum_{j=1}^{N_r} \log(1-\omega_{rjk}) & k \leq K_r \\ 0 & k > K_r \end{cases} \quad \forall r \in \{o,i,m\} \qquad \tilde{t}_{rk} = \begin{cases} \sum_{j=1}^{N_r} \omega_{rjk} & k \leq K_r \\ 0 & k > K_r \end{cases} \quad \forall r \in \{o,i,m\}$$

$$\tilde{s}_{mk}^\sigma = \begin{cases} \sum_{j=1}^{N_m} \log(1-\omega_{mj\sigma^{-1}(k)}) & k \in \sigma([K_m]) \\ 0 & k \notin \sigma([K_m]) \end{cases} \qquad \tilde{t}_{mk}^\sigma = \begin{cases} \sum_{j=1}^{N_m} \omega_{mj\sigma^{-1}(k)} & k \in \sigma([K_m]) \\ 0 & k \notin \sigma([K_m]) \end{cases} .$$

(30)

and where the symbol $\overset{\sim}{\geq}$ denotes the use of the first-order Taylor series approximation $-\mathbb{E}\left[\log\Gamma\left(|\mathcal{K}|+1\right)\right] \simeq -\log\Gamma\left(\mathbb{E}\left[|\mathcal{K}|\right]+1\right)$. As with the previous two models, $s_k$ and $t_k$ possess an efficient update scheme. However, since the regularization bound does not decompose as a sum over components $k \in [K_i + K'_m]$, the Hungarian algorithm cannot be immediately applied. Investigating approximations for the troublesome concave $-\log\Gamma\left(\cdot\right)$ term is left for future work.