[Reviews · NeurIPS 2015]

Submitted by Assigned_Reviewer_1

Quality and Clarity: this paper is well written and easy to understand. The authors did good work for emphasizing the strong point of the proposing algorithm.

Originality: this paper proposes streaming, distributed variational approximation scheme for BNP models. The main contributions seem to be twofolds; minibatch posterior decomposition that enables a learning rate free streaming distributed computing, and component identification framework to maximize variational posterior. While the first one is not very original since the learning rate free posterior decomposition for variational approximation was already proposed, the second one is quite novel. The most of the existing variational approximation schemes for DP mixtures tried to solve the problem in stick-breaking formulation with truncation, since in that case the computation of expectation over variational distributions are relatively easy than that of CRP-based formulation. In this paper, however, the authors directly tackle the combinatorial problem as a matching problem, and this yields a simple yet useful algorithm which does not require truncation or sums over exponentially many partitions. Also, it is suitable for distributed inference.

Significance: scalable posterior inference algorithms for BNP models have been very important recently, and this work would have a significant impact on the community.

Minor comments:

- it seems that the variational inference procedure for minibatch needs a truncation, even if the overall procedure is truncation free. How do you select the truncation level for minibatches, and how does it affect the computational complexity and accuracy of the matching problem? - the supplementary material includes the regularization bound for Pitman-Yor mixtures, and it would be good to see the properties of that bound compared to DP bounds. - the subscript 0 (zero) and o (original) is quite misleading.
Summary: The problem that the paper tries to solve is important for BNP communities Well written paper with good supporting experiments

Submitted by Assigned_Reviewer_2

The paper takes advantage of a model density decomposition due to Bayes' theorem to design a distributed variational inference algorithm.

Practical experiments are performed with the DP-mixture model.

The experiments section is terrific, and on those grounds alone I vote to accept the paper.

To use the DP-mixture model on these datasets with a practical mind towards setting the problem up on a non-trivial computing infrastructure is something lacking in the current Bayesian nonparametrics output, so I really appreciate this contribution.

By the way, the "streaming" concept was a bit misleading to me.

I thought the streaming inference would be central to the application, but as far as I can tell, it is only an aspect of the inference procedure.

In this sense, aren't all Bayesian nonparametric models already streaming?

So why the focus on this?

I think I may be missing something...

As a minor comment, the paper's presentation rubs me the wrong way a bit.

Unless I'm missing something, in my opinion, there are no new ideas here.

The decomposition in Eq (1), the approximation in Eq. (3), and the distributed nature of such Bayesian models are all obvious, and such properties are what make these models so attractive in the first place, because they could in theory be used for distributed and "streaming" inference.

(Admittedly, I don't have experience with how serious of an issue the component matching in Sec 2.2 is.)

I agree that existing papers DON'T do all of this, though, and for this reason the paper is very welcome.

I would have just preferred if the paper didn't sound like it was claiming to be introducing a "novel framework", instead, it's just dealing with the so-far ignored issues to make all of these Bayesian nonparametric models practical.

Just my two cents.
Summary: The paper sets up distributed inference algorithm which perform variational inference with minibatches.

The DP-mixture model is applied to practically sized problems to good effect.

Submitted by Assigned_Reviewer_3

Review Summary ============== Score: 6, Marginally above the acceptance threshold

The proposed method for streaming, distributed inference of DP mixture models presents a nice solution to the cluster identification problem, backed by experiments that are convincing though not rock solid. I'm hesitant to recommend unconditional acceptance, because basic information about how new clusters are created at each minibatch are totally absent, hurting reproducibility.

Summary of Paper ================

This paper develops a new algorithm for streaming, distributed variational inference for the DP mixture model, with some supplementary material suggesting how to use these insights for many other BNP models. Using a mean-field approximation, the authors consider how to allow multiple worker nodes to process data batches in parallel and then aggregate these results asynchronously. In particular, the authors offer a new solution to the "component identification" problem: how to find correspondence between new clusters created independently by two separate worker nodes. This solution finds an optimal assignment for each new cluster from the current minibatch, merging it with another new cluster if advantageous but keeping it separate otherwise. The optimization objective for this assignment is derived via applying Jensen's inequality to the exchangeable partition probability function (EPPF), which works out nicely for the DP case and applies to some other BNP models.

Experiments compare the proposed algorithm in detail to several other methods for DP mixture models.

Toy data experiments show that with the proposed component identification scheme, they see big speedup using many workers (up to 48) without loss in predictive performance.

Furthermore, the authors report computation time and test log likelihood on 3 "real-world" datasets, showing their method achieves among the best testLL scores in much less time, often with 10x-100x speedup over other methods.

Technical Comments on Method ============================

My biggest concern here is that the paper lacks details about what exactly happens at each minibatch, and especially how exactly new clusters are created for that minibatch.

Line 356 says the proposed approach is coupled with "standard variational inference" (Blei & Jordan 2006), but this is a well-known fixed truncation method and does not create new clusters at all.

It seems the proposed component identification method applies regardless of the specific creation procedure used, but not specifying any details about creation is serious cause for concern. In my experience, the single most important feature of a BNP inference method is how successfully it creates high-quality new clusters that are missing from the current model.

A related concern is that the presented merge scheme in Sec. 3.2 only applies to merging *new* clusters with other new clusters (created by different nodes). Many existing papers highlight ways to merge or remove existing clusters in approximate inference [eg Lin's ref 7, Bryant & Sudderth ref 11]. Without these moves, in my experience junky new clusters can overwhelm good ones. The proposed method does not handle this at all, and so likely underperforms.

A final concern is that the current exposition does not compare and contrast the proposed DP mixture regularization bound (Eq 9) with existing mean-field variational objectives (e.g. from Blei and Jordan 2006 or Hughes & Sudderth 2013). Adding some of these bounds to Fig 2 would be interesting. It's unclear whether the proposed bound leads to a coherent overall bound on the marginal likelihood... I think so but it should be stated more obviously. It's also unclear even what objective function is used within a minibatch for SDA-DP: the proposed one? or the one from Blei and Jordan 2006?.

Technical Comments on Experiments =================================

The toy data results (Fig 3) are pretty convincing, though if you have 2D data showing plots of learned cluster parameters overlaid with data is nice. My big question here is that from Fig 3e it seems most runs do not recover all K=100 true clusters, instead usually about 90 or 95. This screams local optima to me, and I'm curious whether this might go away with a longer run (if so, how long would you need to run?), or whether the algorithm does really gets stuck if even run for a while and fails to create these missing clusters. Finally, figure 3b shows no visible speedup from 24 to 48 workers... is this because the problem is too small for effective parallelization? Or due to overhead from locking/unlocking?

I'm less convinced by the experiments in Fig. 4. First, please plot traces of algorithm performance, rather than using a table, to communicate more information in the same amount of space. Second, lots of experimental details are missing: how was prior set? how were competitor methods initialized? what final number of clusters is reached for each dataset? was just one run used, or the best of multiple runs of each method? did you do many passes through the dataset, or just one pass? Third, some of the Airplane testLL numbers are so low (esp. for SVA and SC) that it's hard to know whether to blame local optima, buggy code, or what, but I'd expect both of these to perform better.

Overall, the lack of detail makes these experiments hard to evaluate and hard to reproduce. It's also unclear how performance is sensitive to batch size, and whether competitor algorithms were allowed to vary the truncation... did the SVA runs use that paper's merge/prune moves? did the moVB runs include that paper's birth and merge moves? did SVI and Batch just use fixed truncation?

Novelty ======= Other work provides the basic framework for mean-field inference [ref 24, etc.] and streaming/asynchronous Bayesian computation [ref 5, etc] used here. The real contribution is the new framing of the "component identification" problem, which I have not seen adequately addressed in prior work, and especially the concrete bound for the DP mixture case.

Significance ============ The problem of general-purpose scalable inference for BNP models is definitely worth studying. The proposed approach could likely be adopted by others considering that it is based on the EPPF.

Clarity =======

I could follow the main arguments without too much trouble, but in many places (like around Eq 10) the authors relied too heavily on math notation and lacked motivating narrative. Around Eq. 10, I'd recommend adding text offering interpretations of the statistics s and t used for computing the regularization bound.

Another possible improvement: In figure 1, remove the nodes for y_t illustrating data batches (which dont add much) and instead make the focus on the cluster parameters. Perhaps show two different cycles of a,b,c,d, so the reader better understands how the number of cluster parameters grows over time and what the "component identification" problem is ... maybe highlight the edges that need to be optimized in c from those that are known in advance.

Line-by-line Comments ===================== (not necessary to respond to these in rebuttal)

Lines 51-52: ref 13 does not seem to require a learning rate

Fig 2a: The "U"-shaped bend with large alpha is real but not very intuitive. Could use some commentary. Also, labeling some specific alpha values would help interpret the plot.

Line 289: Saying "the bound increases as K -> 0" might suggest to non-experts that for any N, the DP's preferred number of clusters is very small, which is definitely not true.
Summary: The proposed method for streaming, distributed inference of DP mixture models presents a nice solution to the cluster identification problem, backed by experiments that are convincing though not rock solid. I'm hesitant to recommend unconditional acceptance, because basic information about how new clusters are created at each minibatch are totally absent, hurting reproducibility.

Author Feedback
Author rebuttal: We thank the reviewers for their time and effort spent assessing our manuscript, and for the generally positive feedback. Reviewers are addressed individually below.

R1: As with most variational methods, the truncation for minibatch inference must be selected heuristically. However, note that using a truncation-free method for minibatch inference avoids this issue (such methods are ideal to use within our framework for this very reason). The matching problem is unaffected by empty clusters (they can be trivially pruned before optimization), so setting a large truncation on each minibatch is a good idea, modulo the problem-specific cost of minibatch inference. We suggest increasing the truncation by K_inc beyond the number of clusters in the "original" posterior for each minibatch step.

R2: Indeed, BNPs are "streaming priors" (they express expected model growth with increasing amounts of data). However, during inference, exchangeability causes a dense dependence graph between observations, making streaming inference difficult. This is where our paper excels -- we provide a streaming inference procedure, to match the natural "streaming" capability of BNPs. Actually, the key contribution in this regard *is* indeed the component matching (as noted by R1). Our methodology allows us to break certain dependencies to make streaming inference possible, and then fix these broken dependencies approximately via matching during merge.

R3: We appreciate your careful review of the manuscript; many thanks! First, as noted by R1, even though each minibatch procedure *may be* truncated (depending on the variational algorithm employed), the overall procedure isn't. To perform inference in each minibatch, a truncation level K is selected beyond the set of "old clusters", so that the model can expand. Then when the component matching is solved, these "new" components are matched to one another. Although we used the Blei/Jordan '06 algorithm in the experiments, we could have used any black-box variational algorithm; we just wanted to make sure that our framework was responsible for any performance increases, rather than a "better" underlying inference algorithm. To help with clarity, we will add remarks to this effect to the beginning of Section 2.2.

Next, we agree that prune/merge at the central node would be a great idea, and are actively pursuing its development. However, we believe the paper is strong enough for publication without this material.

Our bound solves a different problem than standard variational bounds, and so the two should not be compared. While mean-field bounds the true marginal likelihood, ours simply bounds the marginal likelihood *assuming the variational posteriors are correct*. Note that the proposed objective (Eqs 5, 12, 13) is only applied when solving the component matching problem. During minibatch inference, the variational bound (e.g. Blei/Jordan '06) is used. Our to-be-added remarks at the beginning of 2.2 will clarify this.

Our algorithm, as with all variational methods, is susceptible to local optima. The prune/merge discussed earlier will help with this, but as mentioned, this is left for future work. Figure 3b shows no performance improvement beyond 24 threads because our computer had 24 cores; we just wanted to show that additional parallelism didn't hurt the solution quality. We will mention the number of CPU cores in this section.

For the experiments: Priors were set by hand tuning for the best performance. All methods were initialized randomly. SDA-DP, by design, only does one pass through the data, but any algorithm that could use multiple passes (moVB, SVI, etc) did so. We used implementations of both SVA and SC provided by the original authors; SC had trouble converging no matter how we started it, and SVA either created lots of junky clusters or dumped everything in one big cluster, despite using prune/merge moves (it is very sensitive to its tuning parameters). moVB was allowed to make birth/death moves. SVI/Batch had a fixed truncation. We will include this information in the final draft/supplement.

R4: Different batches are distributed to different nodes and processed concurrently. Any time a new mini batch "arrives", it is dispatched to a waiting node and processed there asynchronously. We will try to clarify this in the final draft by removing "node B"and "node C" from Figure 1, which could cause confusion. Comparing our algorithm with others vs the amount of observed data may not be a good idea, since our algorithm is constrained to only see data once. It wouldn't make sense to artificially modify (and break) other algorithms to make sure they only see data once for comparison purposes. In the other direction, it may be possible to extend our algorithm to allow multiple views of the data (if computation resources are otherwise idle), but that is significant enough of a change that we leave it for future work.

R5&6: Thanks for the positive feedback!